# The Antimicrobial Activity of Peripheral Blood Neutrophils Is Altered in Patients with Primary Ciliary Dyskinesia

**DOI:** 10.3390/ijms22126172

**Published:** 2021-06-08

**Authors:** Maaike Cockx, Marfa Blanter, Mieke Gouwy, Pieter Ruytinx, Sara Abouelasrar Salama, Sofie Knoops, Noëmie Pörtner, Lotte Vanbrabant, Natalie Lorent, Mieke Boon, Sofie Struyf

**Affiliations:** 1Laboratory of Molecular Immunology, Department of Microbiology, Immunology and Transplantation, Rega Institute for Medical Research, University of Leuven, 3000 Leuven, Belgium; maaike.cockx@kuleuven.be (M.C.); marfa.blanter@kuleuven.be (M.B.); mieke.gouwy@kuleuven.be (M.G.); pieter.ruytinx@kuleuven.be (P.R.); sarah.abuelasrar@kuleuven.be (S.A.S.); sofie.knoops@kuleuven.be (S.K.); noemie.portner@kuleuven.be (N.P.); lotte.vanbrabant@kuleuven.be (L.V.); 2Pneumology and Cystic Fibrosis Unit, Department of Pneumology, University Hospitals Leuven, 3000 Leuven, Belgium; natalie.lorent@uzleuven.be; 3Pediatric Pneumology and Cystic Fibrosis Unit, Department of Pediatrics, University Hospitals Leuven, 3000 Leuven, Belgium; mieke.boon@uzleuven.be

**Keywords:** primary ciliary dyskinesia, neutrophils, antibacterial activity

## Abstract

The airways of patients with primary ciliary dyskinesia (PCD) contain persistently elevated neutrophil numbers and CXCL8 levels. Despite their abundance, neutrophils fail to clear the airways from bacterial infections. We investigated whether neutrophil functions are altered in patients with PCD. Neutrophils from patients and healthy controls (HC) were isolated from peripheral blood and exposed to various bacterial stimuli or cytokines. Neutrophils from patients with PCD were less responsive to low levels of fMLF in three different chemotaxis assays (*p* < 0.05), but expression of the fMLF receptors was unaltered. PCD neutrophils showed normal phagocytic function and expression of adhesion molecules. However, PCD neutrophils produced less reactive oxygen species upon stimulation with bacterial products or cytokines compared to HC neutrophils (*p* < 0.05). Finally, the capacity to release DNA, as observed during neutrophil extracellular trap formation, seemed to be reduced in patients with PCD compared to HC (*p* = 0.066). These results suggest that peripheral blood neutrophils from patients with PCD, in contrast to those of patients with cystic fibrosis or COPD, do not show features of over-activation, neither on baseline nor after stimulation. If these findings extend to lung-resident neutrophils, the reduced neutrophil activity could possibly contribute to the recurrent respiratory infections in patients with PCD.

## 1. Introduction

Primary Ciliary Dyskinesia (PCD) is a rare, autosomal recessive disorder that is caused by genetic defects in motile cilia, resulting in impaired mucociliary clearance [1,2,3]. Patients with PCD suffer from recurrent upper and lower respiratory tract infections, and in almost 50% of the cases, laterality defects are detected [1,4]. Diagnosis and therapy are challenging due to the rarity (estimated prevalence 1/10,000–1/40,000) and heterogeneity of PCD [5,6,7]. Current treatment strategies for PCD only focus on relieving the symptoms and are mainly adaptations of guidelines for the treatment of patients with more common respiratory diseases, such as cystic fibrosis (CF) [5,7]. Typically, patients remain symptomatic and overall experience a decreased quality of life [5,8]. 

Apart from dysfunctional cilia, there are indications that the innate immune system might be affected as well. Indeed, high neutrophil (polymorphonuclear cell, PMN) numbers and CXCL8 levels have been detected in the lungs of patients with PCD, but despite this, persistent bacterial infection and inflammation are frequently observed [9,10]. Other respiratory diseases are also characterized by high PMN influx (e.g. CF, non-CF bronchiectasis, chronic obstructive pulmonary disease (COPD), severe asthma and acute respiratory distress syndrome (ARDS)) [11,12,13]. In these diseases [14], PMNs show abnormal activity and contribute to the lung pathogenesis [15,16]. PMNs are the most abundant leukocytes in the blood and eliminate extracellular pathogens by several mechanisms [17,18]. The first mechanism is phagocytosis, in which microbes are taken up and degraded by the cell. The second mechanism is the release of toxic substances, such as granular proteases and reactive oxygen species (ROS). Yet another mechanism is the expulsion of neutrophil extracellular traps (NETs): strands of DNA mixed with histones and granule proteins, which can immobilize and kill the pathogens [19]. Finally, PMNs can produce cytokines, which attract other immune cells to the site of inflammation. Too few or dysfunctional PMNs cause immunodeficiency, whereas overactive PMNs contribute to excessive inflammation and autoimmunity [20,21,22,23]. In many inflammatory respiratory diseases, increased enzyme release (neutrophil elastase (NE), myeloperoxidase (MPO) and matrix metalloproteinase 9 (MMP-9)), high ROS production and uncontrolled NET formation are often observed in the lungs [11,24,25,26]. Uncontrolled release of these mediators by PMNs causes damage to the lung epithelium and promotes inflammation. The dysfunction of the PMNs is already detectable in the blood circulation, which can partly be explained by the high plasma levels of pro-inflammatory cytokines [11,27,28,29]. Today, interventional trials are running to attenuate PMN activity by targeting their released mediators (e.g., NE) in inflammatory lung diseases [30]. In CF, it has been established that the genetic defect directly affects PMN activity, which is further aggravated by the chronic inflammatory conditions in CF lungs [16]. Understanding the role of neutrophils in PCD is important for the improvement of diagnostic tools and treatment. Until now, only the che-motactic migration of peripheral blood PMNs of patients with PCD has been studied in detail. We previously observed a reduced migration of PMNs from patients with PCD to the chemokines CXCL5 and CXCL8 compared to PMNs from healthy controls, whereas the chemotactic response to leukotriene B4 (LTB4) and complement component 5a (C5a) was not significantly reduced [31]. Other researchers, however, did observe reduced chemotactic migration of PCD PMNs to LTB4 and C5a, as well as to the bacterial product N-formyl-methionyl-leucyl-phenylalanine (fMLF) [32]. Today, both the molecules and pathways involved in leukocyte migration are better characterized and new migration assays have become available, so we re-investigated the PCD PMN response to fMLF to resolve the contradictions.

In view of the high PMN counts and CXCL8 levels in the airways of patients with PCD [9,10], we hypothesized that, similar to the other PMN-dominated respiratory diseases, dysfunctional PMNs play a role in PCD lung pathogenesis. To verify this hypothesis, we performed several activity assays on peripheral blood PMNs from patients with PCD and compared them with PMNs from healthy individuals. To this end, we first isolated PMNs from peripheral blood and tested their chemotactic response to fMLF in the Boyden chamber assay, shape change assay and µ-slide chemotaxis assay to resolve previous contradictory results [33]. Simultaneously, the expression of the corresponding chemoattractant receptors (i.e., FPR1 and FPR2) on PMNs was measured by flow cyto-metry. Furthermore, we measured the expression of several adhesion molecules on PMNs from patients with PCD and healthy individuals to assess their capacity to transmigrate through the endothelium under inflammatory conditions. Finally, several antibacterial activity tests were performed to measure the production of ROS, NET formation and the production of cytokines and chemokines upon PMN stimulation

## 2. Results

### 2.1. Patients and Controls

Between November 2012 and February 2021, 48 patients with PCD, 33 age-matched healthy controls and 21 healthy adult controls were recruited. For more details on the clinical characteristics of the patients, see Table 1.

### 2.2. Peripheral Blood Neutrophils from Patients with PCD Show Reduced Chemotactic Response to fMLF in Three Chemotaxis Assays

We assessed the chemotactic response of PCD PMNs to fMLF in three different assays: the Boyden chamber assay, the shape change assay and the µ-slide chemotaxis assay. PMNs from patients with PCD and healthy controls (both pediatric and adult) showed similar responses to fMLF (10^−9^ M) in the Boyden chamber and the shape change assays (Figure 1A,C). However, we did observe significantly reduced migration of PCD PMNs to the lowest concentration of fMLF (10^−10^ M) compared to PMNs from adult controls (**** *p* < 0.0001 and * *p* = 0.0344; Figure 1B,D, respectively) and pediatric controls (^$^
*p* = 0.0302; Figure 1D) in both chemotaxis assays.

The migratory capacity of PCD PMNs to fMLF was also tested in a more advanced assay, the µ-slide 2D chemotaxis assay. As only a limited number of conditions could be tested in this device, we first optimized the assay to identify the optimal fMLF concentration, which was 10^−8^ M. PMNs did not respond to fMLF 10^−10^ M in this latter chemotaxis assay (data not shown). In control conditions (i.e. buffer), we detected no difference in spontaneous migration between the PMNs of patients and healthy adults. We did, however, observe reduced migratory capacity (objectified by 4 different parameters shown in the different panels of Figure 2, all significantly aberrant) of PCD PMNs to fMLF 10^−8^ M.

Simultaneously with the chemotaxis assays, we measured the expression levels of the fMLF chemoattractant receptors FPR1 and FPR2 on PMNs of the three donor groups by flow cytometry. The expression of these receptors on PMNs was similar between the patients with PCD and both healthy control groups (Figure 3).

### 2.3. Compared to Peripheral Blood Neutrophils of Healthy Individuals, Neutrophils from Patients with PCD Express Similar Adhesion Molecule Levels upon Stimulation

To reach infected tissues, PMNs need to extravasate. Endothelial adherence and transmigration are mediated by successive interactions between adhesion molecules (expressed by both PMNs and endothelial cells), the expression of which is regulated by cytokines and bacterial stimuli [34,35]. In our experimental setup, stimulation with fMLF, TNF-α and PGN resulted in upregulation of the integrin CD11b and/or downregulation of the selectin CD62L. CD11a expression did not change significantly upon stimulation (Figure 4). Furthermore, we found no difference in expression levels of these adhesion molecules on resting PMNs from patients with PCD and either of the healthy control groups (data not shown). When the efficacy of the different stimuli was compared, fMLF stimulation had the most pronounced effect on CD11b upregulation (Figure 4A) and TNF-α on CD62L downregulation (Figure 4C). Overall, the response of PCD PMNs was rather similar to that of pediatric and adult controls. Only CD11b expression appeared to be somewhat higher on PCD PMNs compared to the age-matched pediatric controls after PGN treatment (^$^
*p* = 0.0206) (Figure 4B). In contrast, CD62L seemed to be less readily downregulated on PCD PMNs compared to PMNs from healthy individuals in response to PGN stimulation (Figure 4B). However, due to high variability in the data, this difference was not significant. Finally, adhesion molecule expression levels on PMNs after fMLF or TNF-α stimulation did not differ between the three groups (Figure 4A,C).

### 2.4. The Phagocytic Capacity of Peripheral Blood Neutrophils Is Not Affected in Patients with PCD

To assess phagocytosis function, we measured the uptake of *Staphylococcus aureus*-coated beads by PMNs from patients with PCD and healthy controls. The phagocytic capacity of PCD PMNs was similar to that of PMNs from both healthy control groups (Figure 5).

### 2.5. Peripheral Blood PMNs from Patients with PCD Produce Less Reactive Oxygen Species Compared to PMNs from Age-Matched Pediatric Controls

We measured the production of reactive oxygen species (ROS) by PMNs in response to different bacterial and physiological stimuli. Results (Figure 6) are displayed as the fold change (i.e., the difference between start value and highest value, divided by the start value) of ROS production by PMNs from patients with PCD or pediatric controls relative to the fold change of ROS production by PMNs from the reference adult control. Figure 6A displays a representative graph of ROS production by PMNs from an adult control. The production of ROS in response to a stimulus showed a characteristic curve with the ROS production increasing up to a maximum value and then going down again. Relatively little ROS was produced in response to buffer. It should be noted that under control conditions (i.e. untreated cells not primed with TNF) PMNs from healthy children (median age 8 years) released significantly more ROS (** *p* = 0.0028) compared to PMNs from healthy adults (median age 25 years). Therefore, ROS production by neutrophils from PCD patients should be primarily compared with the pediatric control group, rather than the adult healthy controls. Under control conditions PCD PMNs (median age 8 years) produced less ROS compared to PMNs from Ped CO (^$^
*p* = 0.0125) (Figure 6B). We observed no differences in ROS production between PMNs from patients with PCD and pediatric controls upon PMA stimulation (Figure 6C), but priming with TNF-α resulted in enhanced ROS secretion by PMNs from pediatric controls compared to PCD PMNs (^$$^
*p* = 0.0097) and PMNs from adult controls (**** *p* < 0.0001) (Figure 6D). Furthermore, stimulation with fMLF, LPS and IL-1β after TNF priming resulted in significantly lower ROS production by PCD PMNs compared to PMNs from both control groups (fMLF, ^$^
*p* = 0.0279, ** *p* = 0.0015; LPS, ^$^
*p* = 0.0465, * *p* = 0.0135; IL-1β, ^$$^
*p* = 0.0014, ** *p* = 0.0015) (Figure 6E,F,H). PGN stimulation after TNF-α priming provoked similar ROS production by PMNs from the three donor groups (Figure 6G).

### 2.6. Circulating PMNs from Patients with PCD Tend to Release Less Neutrophil Extracellular Traps Compared to PMNs from Healthy Adults

To investigate whether NET release is disturbed in patients with PCD, PMNs from adult and pediatric patients with PCD and healthy adults and children were stimulated and subsequently monitored by the IncuCyte S3 Live-Cell Analysis System. Every 15 min, pictures of the cells were taken, for a total period of 5 h (Figure 7). This is the only figure wherein more than half of the patients included were adults. In addition, the PMN from healthy children released more NETs than healthy adults. Therefore, we show the results obtained with children and adults separately. Figure 7A shows a representative example of the images at time point 0 and time point 5 h. We found that the fluorescence in most conditions kept increasing up to 5 h and therefore used this time point as the outcome measure. The results in Figure 7B–E are displayed as the area of fluorescence (caused by extracellular DNA reacting with SYTOX Green) relative to the total cell area at timepoint 0 (in %). There seemed to be a trend towards increased NET formation at baseline (i.e. in the absence of an external stimulus) between PMNs from pediatric and adult individuals. In addition, there seemed to be a difference in baseline NET formation between PMNs from patients and healthy controls (Figure 7B). The difference between PMNs from patients with PCD and healthy controls remained when PGN was added to stimulate the release of NETs. The addition of TNF-α to the cells had no significant effect compared to the control. Although at present, the only statistically significant differences were obtained between adult patients with PCD patients and healthy children, a clear trend towards reduced NET formation can be noted (*p* = 0.066).

### 2.7. Peripheral Blood Neutrophils from Patients with PCD Produce Less CXCL8 Compared to Neutrophils from Healthy Children

We measured the cytokine production by PMNs in response to bacterial stimuli. Compared to healthy children, PCD PMNs produced less CXCL8 upon stimulation with fMLF and LPS (Figure 8A,B, respectively), whereas PGN stimulation resulted in similar CXCL8 production (Figure 8C). fMLF did not induce IL-1β production in this experimental setup. Similar levels of IL-1β were produced by patients with PCD and healthy controls upon stimulation with LPS and PGN (Figure 8D,E; *p* = 0.3971 and *p* = 0.3229, respectively).

## 3. Discussion

Primary Ciliary Dyskinesia (PCD) is a genetic disorder that is characterized by recurrent respiratory infections [1,2]. Mutations in several ciliary genes result in structural and functional abnormalities of the motile cilia, which are essential for mucociliary clearance, an important primary defense mechanism in the human airway. Besides defects in mucociliary clearance, there are reasons to believe that also neutrophilic function might be altered in PCD. We previously observed a reduced chemotactic response to the chemokines CXCL5 and CXCL8 for peripheral blood PMNs from patients with PCD. Here, we studied in detail the antimicrobial effector functions of peripheral blood PCD PMNs.

To resolve the contradictions [32,36,37] in the field, we first assessed the chemotactic response of PCD PMNs to the bacterial peptide fMLF in three different assays. Compared to PMNs from healthy children and adults, PCD PMNs showed a similar response to fMLF at the concentration of 10^−9^ M, but a significantly lower response to 10^−10^ M fMLF in the Boyden chamber assay and the shape change assay, suggesting that neutrophils from patients with PCD might be less sensitive to external stimuli. In the 2D µ-slide chemotaxis assay, PMNs from patients with PCD were also significantly less responsive to fMLF compared to PMNs from the reference Ad CO. Similarly, Koh et al. found reduced migratory capacity of PCD PMNs to fMLF in yet another chemotaxis assay [32].

We hypothesized that the reduced chemotactic response to fMLF of PCD PMNs could be related to receptor internalization or desensitization. To test this, the expression of fMLF chemoattractant receptors FPR1 (high affinity) and FPR2 (low affinity) was measured on PMNs from patients with PCD and healthy individuals. We observed no difference in fMLF receptor expression on the surface of PMNs between the three groups. Of note, FPR1 desensitization is not always associated with receptor internalization, as mechanisms other than arrestin binding regulate termination of FPR signaling, for example, direct blockade of G protein binding by the actin cytoskeleton [38]. Thus, our results do not exclude receptor desensitization as a possible cause for reduced response of PCD PMNs to fMLF.

The next feature that we studied in PCD PMNs was their capacity to adhere to the endothelium via analysis of adhesion molecule expression after stimulation with bacterial components (fMLF, PGN) or pro-inflammatory mediators (TNF-α), which are known regulators of integrin and selectin expression on PMNs [34,39,40]. Upon stimulating the cells, we generally observed no difference in adhesion molecule expression between PMNs from patients with PCD, pediatric controls and adult controls. The only exception was an increased expression of CD11b on PCD neutrophils upon stimulation with PGN. However, the observed difference was minimal and as such only limited effects on blood PCD PMN extravasation are expected. As we detected no further difference in adhesion molecule expression on PMNs from patients compared to PMNs from healthy controls in control and inflammatory conditions, our results collectively indicate that peripheral blood PCD PMNs exhibit normal extravasation features. Similarly, research shows that CF PMNs display normal blood endothelium transmigration [41]; however, upon stimulation, CD62L is shed off less compared to healthy controls [42]. In contrast, blood PMNs from patients with COPD and severe asthma show aberrant adhesion molecule expression, both under control and pro-inflammatory conditions, which correlates with the activated state of blood PMNs in these diseases [43,44,45].

We performed several PMN activity assays to investigate the capacity of peripheral blood PCD PMNs to combat bacterial infections, but we observed no significant differences in the phagocytic capacity between PMNs from patients, age-matched pediatric individuals and adult individuals. This indicates that peripheral blood PMNs from patients with PCD show normal abilities to clear pathogens from host tissue via phagocytosis. Of note, in CF, a profound difference in phagocytosis is observed between blood and airway PMNs, suggesting that some neutrophil defects contributing to inflammation are not detectable at peripheral blood level [46]. Hence, in the future, we will study the phenotype and function of airway PMNs of patients with PCD.

The next mechanism that we examined was the production of reactive oxygen species (ROS), which is important for intracellular bacterial killing after phagocytosis and extracellular killing simultaneously with degranulation. We measured significantly less ROS production by PCD PMNs under basal conditions and in response to TNF-α compared to PMNs from healthy children. Similarly, PCD PMNs showed significantly less response to fMLF, LPS and IL-1β compared to PMNs from both control groups. These results indicate that blood PCD PMNs are less responsive to bacterial and pro-inflammatory stimuli compared to blood PMNs from healthy individuals. The reduced capacity of PCD PMNs to produce ROS presumably affects their ability to kill pathogens intracellularly and extracellularly. We hypothesize that this reduced response is either caused by the genetic defect or otherwise by sustained pro-inflammatory conditions. However, it has not been established yet that enhanced levels of pro-inflammatory mediators are present in the blood circulation of patients with PCD.

We observed a trend towards reduced release of DNA in PCD PMNs in all experimental conditions tested. This result was quite surprising, as peripheral blood PMNs from patients with CF show enhanced NET production [25,47]. The reduced capacity to release extracellular DNA is thus likely to hamper PCD PMNs in their antimicrobial activities. More striking is that children (both healthy and PCD) tended to produce more NETs than adults. This finding suggests that NET formation might be age-dependent; however, more research is needed to test this hypothesis.

At the same time, PMNs from patients with PCD produced less CXCL8 in response to LPS and fMLF as compared to healthy pediatric controls. This finding is intriguing, since previous research indicates elevated levels of CXCL8 in the lungs of patients with PCD [9]. It should be noted, however, that only one timepoint was considered in our experimental setup and it could therefore be possible that patients with PCD have a delayed rather than a decreased production of CXCL8. Furthermore, peripheral blood monocytes of patients with PCD produced more CXCL8 in response to inflammatory triggers than monocytes from healthy individuals [31]. After stimulation with PGN (10 µg/mL) PCD mononuclear cells produced 390 ng/mL of CXCL8, whereas the same number of healthy neutrophils produced 247 ng/mL of CXCL8 in response to PGN.

An important question is whether these data are unique for PCD or if they are a general feature of chronic lung inflammation. In this study, we only tested neutrophil function of patients with PCD and healthy controls; however, previous research shows that the function of peripheral blood neutrophils can be aberrant in other chronic lung diseases as well (Table 2). While cystic fibrosis, COPD and asthma all mostly show an overactivation of blood neutrophils, we found the opposite to be the case in PCD. This indicates that the immune profile in PCD is unique and may not respond well to immune suppressors.

Importantly, the results that we obtained are likely not a primary result of the genetic mutations found in PCD, as most of the genes mutated in our cohort are not expressed in neutrophils [58].

## 4. Materials and Methods

### 4.1. Patients and Healthy Controls

Patients with PCD were recruited at the pediatric department of the University Hospital of Leuven and Université Catholique de Louvain. At the time of blood donation, all patients were clinically stable. Diagnosis of PCD was confirmed by several diagnostic tests, according to the gold standard [2,23]. In this study, we included two healthy control groups: an age-matched healthy control group and an adult control group. The adult control group was used as reference, because it was impossible to organize blood donations of healthy children and children with PCD on the same day. Some participants provided samples on more than one day, in which case we averaged the measurements. All patients and healthy adults and children (or their parents) signed informed consent. The Ethical Committee of UZ Leuven approved the study protocol [S57236(ML11095)].

### 4.2. Sample Collection and Processing

Blood samples were collected in EDTA-coated tubes. To isolate PMNs from whole blood samples, we applied density gradient centrifugation [31]. The blood samples were diluted with Dulbecco’s phosphate buffered saline (D-PBS; Lonza, Basel, Switzerland), gently layered on Ficoll-sodium diatrizoate (Lymfoprep, Axis-Shield PoC AS, Dundee, United Kingdom) and centrifuged (400× *g*, 30 min, 20 °C, without break). The bottom layer with PMNs and red blood cells was first diluted with D-PBS and then gently mixed with 6% Dextran (Sigma-Aldrich, St. Louis, MO, USA) in Milli-Q water to induce red blood cell aggregation and sedimentation. After an incubation period of 30 min at 37 °C, PMNs were isolated. We performed two washing steps with D-PBS and lysed residual red blood cells by a hypotonic shock [31]. After two additional washing steps, PMNs (>95% pure) were ready for use.

### 4.3. Reagents

The bacterial tripeptide N-formyl-methionyl-leucyl-phenylalanine (fMLF,) the bacterial membrane components lipopolysaccharide (LPS) and peptidoglycan (PGN), phorbol 12-myristate 13-acetate (PMA) and luminol (97% purity) were obtained from Sigma Aldrich. The cytokines interleukin 1β (IL-1β) and tumor necrosis factor α (TNF-α) were bought from Peprotech (Rocky Hill, NJ, USA). The antibodies for flow cytometry anti-CD11a (FITC-labeled; mouse anti-human, clone HI111), anti-CD11b (PE-CY5-labeled; mouse anti-human, clone ICRF44), anti-CD16 (PE-labeled; mouse anti-human, clone 555407) and anti-FPR1 (mouse anti-human, clone 556015) were purchased from BD Biosciences (Franklin Lakes, NJ, USA) and anti-CD62L (APC-labeled; mouse anti-human, clone DREG56) from eBioscience, anti-CD16 (APC-labeled; mouse anti-human, clone 3G8) from Biolegend (San Diego, CA, USA), while anti-FPR2 (mouse anti-human, sc-57141) from Santa Cruz (Dallas, TX, USA). The fluorescent probe SYTOX Green was purchased from Thermo Fisher Scientific (Waltham, MA, USA).

### 4.4. Boyden Chamber Chemotaxis Assay

The chemotactic migration of PMNs from patients and healthy controls was tested in the Boyden microchamber chemotaxis assay (Neuro Probe, Gaithersburg, MD, USA), as described before [59]. In short, freshly isolated PMNs from patients with PCD or healthy controls were loaded into the upper wells of a 48-wells Boyden chamber, while the lower wells contained fMLF. After a 45 min incubation, the cells were fixed and stained. The number of migrated cells, attached to the lower side of the membrane, was counted microscopically. The chemotactic index (CI) was calculated by dividing the number of migrated cells to fMLF by the number of spontaneously migrated cells. To compare CIs between Boyden chambers on different days and to compare CIs between patients and pediatric controls, the CI of the adult control tested on the same day was used as reference CI.

### 4.5. Shape Change Assay

A shape change assay was performed as described before [60]. In short, PMNs were shortly (1 min) exposed to fMLF (10^−9^ M and 10^−10^ M). Subsequently, the cells were fixed and their shape was evaluated microscopically (200× magnification) by two researchers, with at least one blinded for the experimental conditions. The percentage of activated cells (irregular cell shape with cellular extensions) and inactive cells (perfectly round cell shape) was determined.

### 4.6. µ-Slide 2D Chemotaxis Assay

µ-slide chemotaxis assay was performed as described before [31]. In short, PMNs from patients with PCD and healthy adults were injected into a µ-slide 2D chemotaxis chamber (IBIDI, Gräfelfing, Germany). To allow PMN adherence, the chamber was incubated for 30 min at 37 °C. Perpendicularly to the PMN-filled channel, fMLF (10^−8^ M) was injected on one side to create a stable concentration gradient. With an inverted microscope (10× phase-contrast objective; Zeiss Axiovert 200 M), a picture of the PMNs was taken every 1.5 min for a period of 2 h. A manual plug-in chemotaxis tool of ImageJ was used to analyze these movies. Twenty randomly chosen cells of each condition were followed throughout the imaging and the average distance, velocity and directionality of the PMNs were calculated.

### 4.7. Flow Cytometry

To determine the expression level of the fMLF receptors FPR1 and FPR2, isolated PMNs were diluted to a concentration of 3 × 10^6^ cells/mL in staining buffer (D-PBS + 2% fetal calf serum) and added to a 96-well plate. After addition of anti-FPR1, anti-FPR2 or PE-labeled anti-CD16 antibody, the cells were incubated for 30 min on ice in the dark. Subsequently, cells were washed three times with staining buffer, and the cells incubated with unlabeled antibodies were labeled with PE-labeled goat anti-mouse antibody and again incubated for 30 min on ice in the dark. The cells were washed 3 times with staining buffer and finally fixed with fixation buffer (staining buffer + 0.4% formaldehyde). FPR1 and FPR2 expression was measured with an FACSCalibur flow cytometer (BD Biosciences). The results were analyzed by CellQuest software (BD Biosciences).

### 4.8. Adhesion Molecule Activity Assay

An adhesion molecule activity assay was performed as previously described [37]. In short, whole blood samples of patients and healthy individuals were stimulated at 37 °C with fMLF (10^−8^ M, 10 min), TNF-α (50 ng/mL, 15 min) or PGN (10 µg/mL, 15 min). After incubation, the cells were immediately placed on ice and stained for the surface molecules CD11a, CD11b, CD16 and CD62L. Subsequently, the cells were fixed and the red blood cells were lysed. Analysis of the adhesion molecules was done by flow cytometry (*vide supra*).

### 4.9. Phagocytosis Assay

PMNs were diluted in phagocytosis uptake buffer (1.5 × 10^6^ cells/mL; Live cell Imaging solution, Molecular Probes, Eugene, OR, USA), whereupon pH Rodo Staphylococcus aureus-coated beads (Molecular Probes) were added to the cells. These beads are sensitive to environmental pH, and their fluorescent signal increases when taken up in the acid phagolysosomes of PMNs. To promote phagocytosis, the cells were incubated at 37 °C for 30 min, while the negative control cells were incubated at 4 °C. Afterwards, both cell suspensions were stained with the APC-labeled monoclonal anti-CD16 antibody for 15 min at 37 °C (negative controls at 4 °C). The cells were then washed twice with staining buffer, fixed and analyzed by flow cytometry (*vide supra*).

### 4.10. Respiratory Burst Assay

To quantify ROS production, PMNs were diluted in RPMI-1640 medium without phenol red (3 × 10^6^ c/mL) and first primed with TNF-α (50 ng/mL) for 10 min at 37 °C. We then transferred the cells to a white 96-well microplate (PerkinElmer, Waltham, MA, USA) and added PMA (30 nM), IL-1β (500 ng/mL), LPS (10 µg/mL), PGN (10 µg/mL) or fMLF (10^−7^ M). Buffer was used to simulate control conditions whereas PMA was used as positive control. Each condition was included in duplicate. The cells were supplemented with 10 mM luminol. The luminol oxidation was monitored for 3 h, measuring the light signal every minute (Clariostar Monochromator Microplate Reader, BMG Labtech, Ortenberg, Germany). During this period, the temperature was kept constant at 37 °C. ROS production by PMNs was assessed by calculating the difference between the minimal and maximal light signal in response to a stimulus, divided by the minimal light signal.

### 4.11. Neutrophil Extracellular Trap (NET) Assay

A 96-well fluorescence plate (Greiner Bio-one, Kremsmünster, Austria) was coated with 100 µL poly-l-lysine per well (100 µg/mL, Sigma Aldrich) for 30 min at RT. PMNs were isolated using an EasySep neutrophil isolation kit (Stemcell Technologies, Vancouver, BC, Canada), following the manufacturers’ instructions. We preferred this method to test NET formation, as the protocol is much shorter compared to the isolation by density centrifugation and yielded better results in this assay. After purification, PMNs were diluted in NETosis buffer (0.5 × 10^6^ c/mL; RPMI 1640 without phenol red + 50 nM SYTOX Green), and 90 µL per well was added to the fluorescence plate. The plate was incubated for 30 min at 37 °C to allow PMN adherence. Afterwards, the cells were stimulated with PMA (150 ng/mL), TNF-α (50 ng/mL) or PGN (5 µg/mL) and placed in the IncuCyte S3 Live-Cell Analysis System (37 °C, 5% CO_2_; Essen BioScience, Ann Arbor, MI, USA). For a period of 5 h, 4 pictures per condition were taken every 15 min with a 20X objective. The fluorescent dye SYTOX Green only binds extracellular DNA, so that the NET formation could be studied over time by measuring the increase in fluorescence. The images were processed by the basic analyzer unit of the IncuCyte S3 2017 Software (Essen BioScience). To measure the total cell area of PMNs in each well of the 96-well plate, the following settings were applied in the ‘Phase Channel’: Segmentation adjustment 0.2; Hole Fill 1; Adjust size +2 pixels; Minimal area 100 µm^2^; Maximal eccentricity 0.99. The total area covered by DNA released by PMNs in each well was determined by applying the following settings in the ‘Green channel’ (wherein fluorescence of DNA-bound-SYTOX Green is detected): Top-Hat segmentation with Radius 100 and Threshold 1.4; Hole fill 0; Adjust size +1 pixel; Minimal area 100 µm^2^; Maximal eccentricity 0.95; Minimal integrated intensity 500; Minimal mean intensity 3. By using an area threshold of 100 µm^2^, we aimed to exclude apoptotic cells (which also stain positive with SYTOX Green), since these are smaller than cells undergoing NETosis [61]. The Edge split tool was turned off in all channels. To calculate the proportion of PMNs producing NETs, the total fluorescence area (green channel) was divided by the total cell area (phase channel) at time point 0 and expressed as a percentage.

### 4.12. Cell Induction

PMNs from patients and healthy individuals were suspended in induction medium (RPMI + 10% FBS + 0.01% gentamycine + 20 ng/mL GM-CSF) at a concentration of 1.5 × 10^6^ cells/mL. GM-CSF was added to prolong PMN survival. The cells were seeded in a 48-well plate and stimulated with fMLF (10^−7^ M), LPS (10 µg/mL) or PGN (10 µg/mL) for 24 h at 37 °C and 5% CO_2_. The supernatants were collected by centrifugation and preserved at −20 °C. We determined levels of CXCL8 and IL-1β in the supernatants by sandwich ELISA [62].

### 4.13. Statistics

For all statistical analyses, we used GraphPad software (GraphPad Software Inc., La Jolla, CA, USA). Normal distribution of the data was assessed by the D’Agostino & Pearson normality test. Since the data were not normally distributed, we applied non-parametric statistical tests. First, non-parametric one-way ANOVA (Kruskal–Wallis test) was performed and afterwards pairwise comparisons (Mann–Whitney U test) to detect statistical differences between two donor groups. In graphs where the results from the reference adult control and patient are compared per experiment, the non-parametric paired Wilcoxon test was applied. A *p*-value < 0.05 was considered statistically significant.

## 5. Conclusions

To conclude, our results signify that peripheral PMNs from patients with PCD do not show features of over-activation, neither under basal conditions nor after stimulation. PCD PMNs rather showed decreased or delayed responsivity to pro-inflammatory and bacterial products. If these findings extend to lung-resident PMNs, the reduced PMN activity could possibly add to the susceptibility of patients with PCD to respiratory infections.

## Figures and Tables

**Figure 1 ijms-22-06172-f001:**
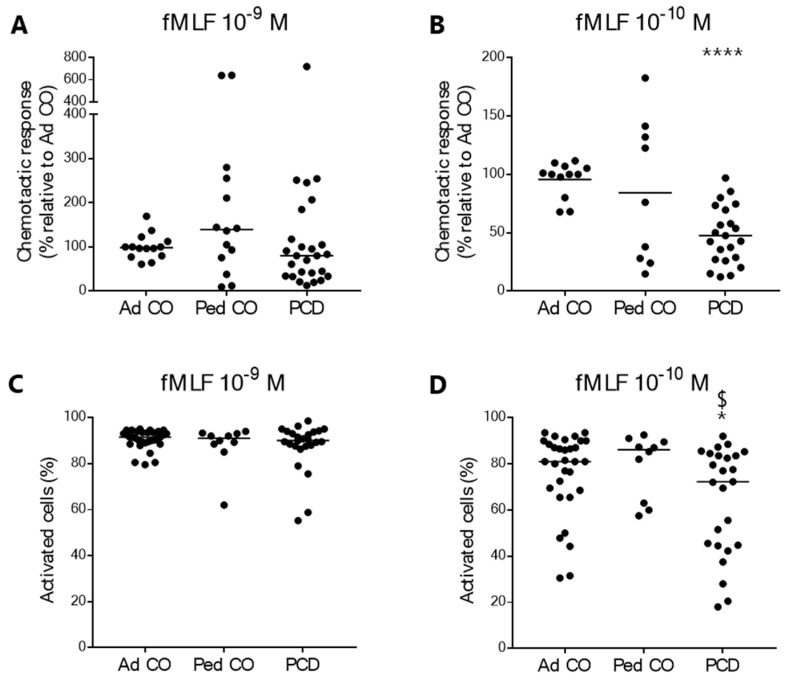
Peripheral blood PMNs from patients with PCD show reduced chemotactic response to low concentrations of fMLF. Chemotactic responses of neutrophils (PMNs) from patients with PCD (PCD, *n* = 22–26 [18–20 pediatric and 4–6 adult]), age-matched healthy pediatric controls (Ped CO, *n* = 9–14) and healthy adult controls (Ad CO, *n* = 12–31) in (**A**,**B**) the Boyden chamber chemotaxis assay and (**C**,**D**) the shape change assay. The chemotactic response to two different concentrations of fMLF was assessed: (**A**,**C**) 10^−9^ M and (**B**,**D**) 10^−10^ M. The chemotactic index (CI; average number of migrated cells in response to the chemoattractant divided by the average number of spontaneously migrated cells) was used to express the chemotactic response. The CIs were normalized to the CI of the corresponding reference Ad CO (%). Statistical differences were determined by Mann-Whitney U-test (Ad CO *versus* PCD: * *p* < 0.05, **** *p* < 0.0001; Ped CO *versus* PCD: ^$^
*p* < 0.05).

**Figure 2 ijms-22-06172-f002:**
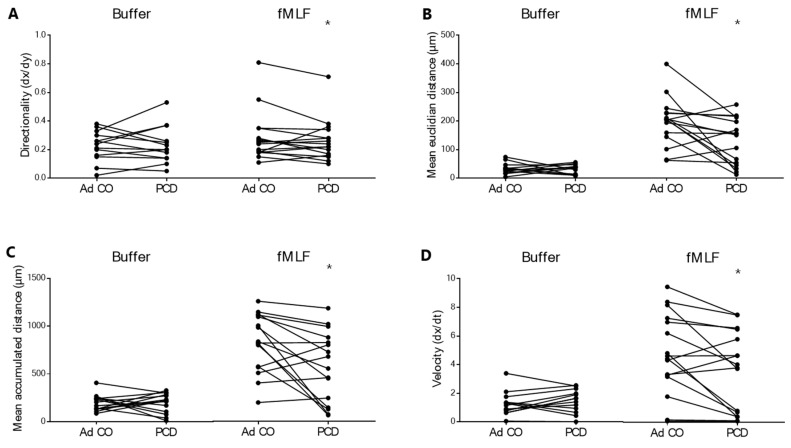
Decreased migratory capacity of blood PMNs from patients with PCD to fMLF in the µ-slide 2D chemotaxis assay. The chemotactic migration of PMNs from patients with PCD (PCD, *n* = 16 [10 pediatric and 6 adult]) and healthy adult controls (Ad CO, *n* = 16) to fMLF (10^−8^ M) was investigated. (**A**) The directionality, (**B**) mean Euclidian distance (µm), (**C**) mean accumulated distance (µm) and (**D**) velocity of PMNs from patients and Ad CO in response to buffer and fMLF were measured. For each experiment, the data obtained from the patient and the adult control tested on the same day are connected by a solid line. Statistical differences were determined by the non-parametrical pairwise Wilcoxon test (* *p* < 0.05).

**Figure 3 ijms-22-06172-f003:**
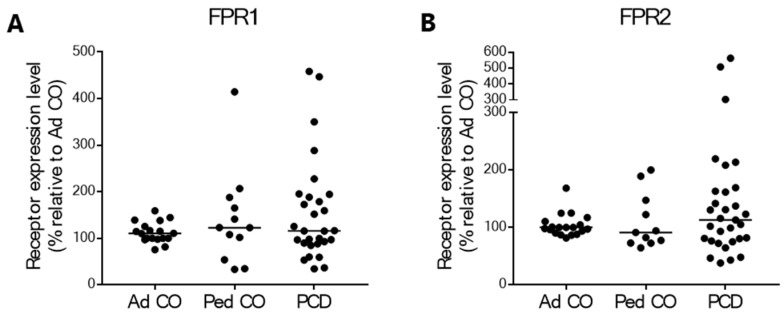
The chemoattractant receptors FPR1 and FPR2 are similarly expressed on peripheral blood PMNs from patients and healthy controls. The expression of fMLF receptors (**A**) FPR1 and (**B**) FPR2 on PMNs from patients with PCD (*n* = 29–31 [21–23 pediatric and 8 adult]), age-matched healthy pediatric controls (Ped CO, *n* =11–12) and healthy adult controls (Ad CO, *n* = 18–19) was assessed by flow cytometry. The chemoattractant receptor expression levels (mean fluorescence intensity, MFI) were normalized to the levels on PMNs of the reference Ad CO (%). Each dot represents the normalized MFI of one patient or healthy control. No statistical differences in chemoattractant receptor levels on PMNs from patients and healthy controls were detected (Mann Whitney U-test).

**Figure 4 ijms-22-06172-f004:**
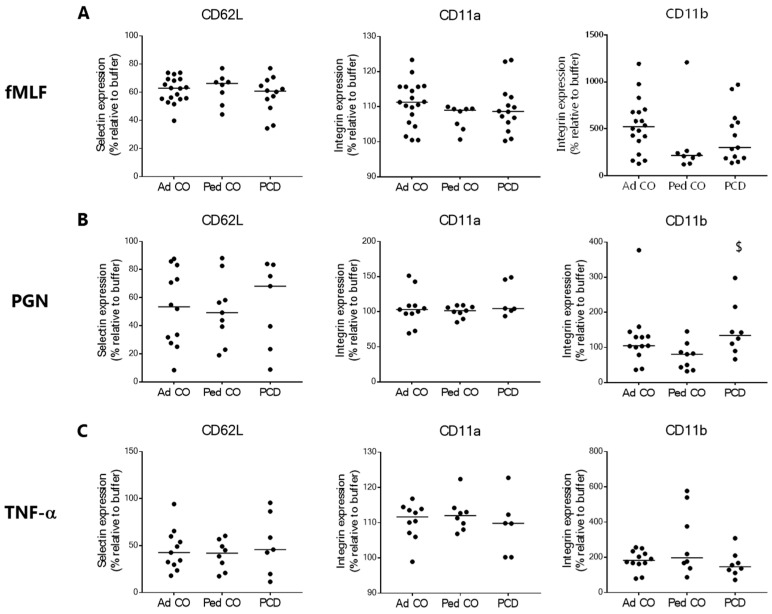
Circulating PMNs from patients with PCD show similar expression levels of adhesion molecules compared to PMNs from healthy controls upon pro-inflammatory stimulation. The differential expression of the selectin CD62L and integrins CD11a and CD11b upon stimulation with (**A**) fMLF (10^−8^ M), (**B**) PGN (10 µg/mL) and (**C**) TNF-α (50 ng/mL) on PMNs from patients with PCD (PCD, *n* = 6–14 [6–14 pediatric and 0–3 adult]), age-matched healthy pediatric controls (Ped CO, *n* = 8–9) and healthy adult controls (Ad CO, *n* = 10–19) was measured by flow cytometry. The expression levels were normalized to the expression level without stimulation (i.e., buffer treatment). Statistical differences were determined by the Mann Whitney U-test (Ped CO *versus* PCD: ^$^
*p* < 0.05).

**Figure 5 ijms-22-06172-f005:**
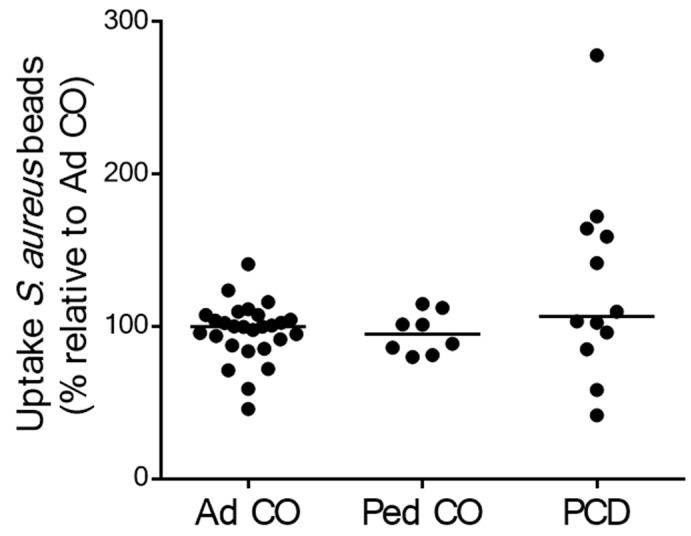
The phagocytic capacity of blood PMNs from patients with PCD is not significantly different from that of PMNs from healthy controls. The phagocytic capacity of PMNs from patients with PCD (PCD, *n* = 12 [10 pediatric and 2 adult]), age-matched healthy pediatric controls (Ped CO, *n* = 8) and healthy adult controls (Ad CO, *n* = 27) was measured by flow cytometry. The uptake of *S. aureus*-labeled beads by PMNs isolated from healthy children and patients was expressed relative to the uptake by PMNs isolated from the corresponding reference Ad CO (%). No significant difference in phagocytosis was detected between PMNs from patients with PCD and either of the healthy control groups (Mann Whitney U-test).

**Figure 6 ijms-22-06172-f006:**
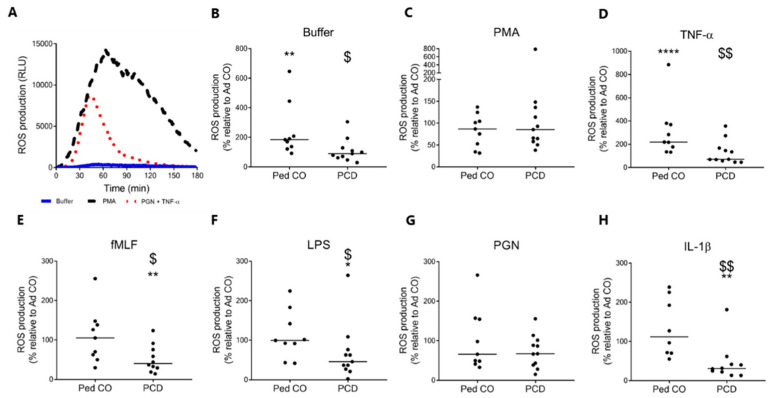
Circulating PMNs from patients with PCD produce less reactive oxygen species compared to PMNs from healthy controls. The production of reactive oxygen species (ROS) by PMNs isolated from patients with PCD (PCD, *n* = 9–11 [7–9 pediatric and 2 adult]), age-matched healthy pediatric controls (Ped CO, *n* = 8–9) and healthy adult controls (Ad CO, *n* = 9–11) was measured by chemiluminescence for 3 h. (**A**) A representative graph of ROS production by PMNs from an Ad CO (relative light units, RLU). ROS production was measured after stimulation with (**B**) buffer or (**C**) PMA (30 nM), which was considered as positive control. Prior to stimulation, PMNs were primed with (**D**) TNF-ɑ (50 ng/mL) for 10 min at 37 °C. Afterwards, PMNs were stimulated with (**E**) fMLF (10^−7^ M), (**F**) LPS (10 µg/mL), (**G**) PGN (10 µg/mL) or (**H**) IL-1β (500 ng/mL). Each dot represents the fold change in ROS production by PMNs from patients and Ped CO relative to the fold change in ROS production of the reference Ad CO (%). Statistical differences were determined by the Mann Whitney U-test (Ad CO *versus* PCD or Ped CO: * *p* < 0.05, ** *p* < 0.01, **** *p* < 0.0001; Ped CO *versus* PCD: ^$^
*p* < 0.05, ^$$^
*p* < 0.01).

**Figure 7 ijms-22-06172-f007:**
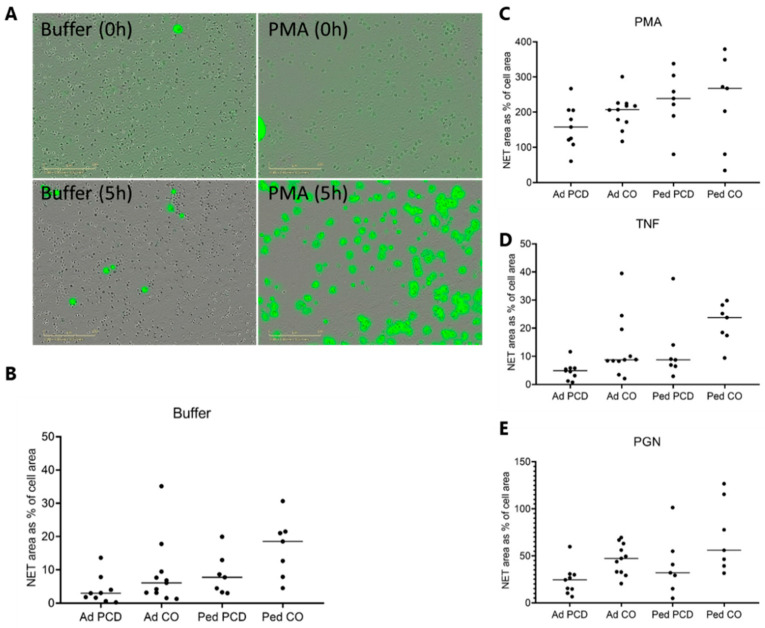
The capacity of PCD PMNs to produce neutrophil extracellular traps is reduced. PMNs from adult patients with PCD (Ad PCD, *n* = 9), healthy adult controls (Ad CO, *n* = 11), pediatric patients with PCD (Ped PCD, *n* = 7) and pediatric controls (Ped CO, *n* = 7) were stimulated with different compounds and incubated for 5 h in the IncuCyte S3 Live-Cell Analysis System. The fluorescent probe SYTOX Green was used to detect the presence of extracellular DNA. (**A**) A representative set of pictures taken by the IncuCyte shows that the fluorescence increases after 5 h of incubation with buffer or PMA. The cells were stimulated with (**B**) buffer, (**C**) PMA (150 ng/mL), (**D**) TNF (50 ng/mL) or (**E**) PGN (10 µg/mL). The production of NETs upon stimulation is displayed as total area of fluorescence relative to the total cell area at 0 h (in %). Statistical differences were determined by the Kruskal-Wallis test. Subsequent Dunn’s multiple comparison test did not indicate significant differences between patients with PCD and healthy controls.

**Figure 8 ijms-22-06172-f008:**
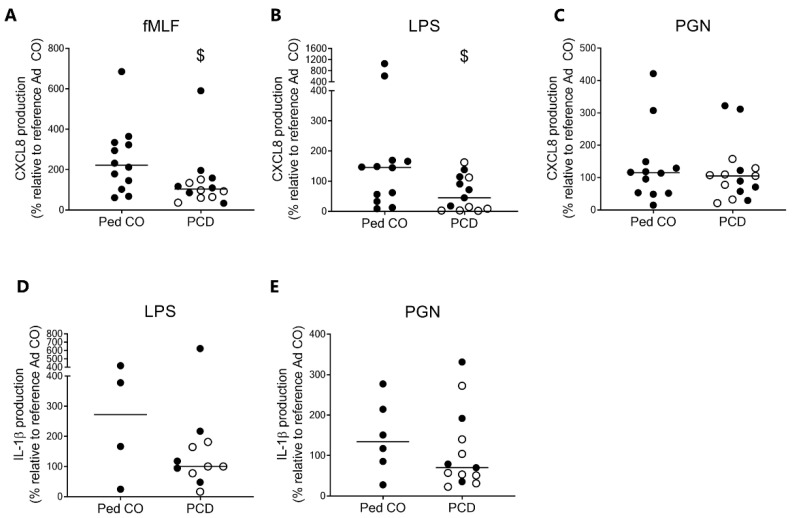
Peripheral blood PMNs from patients with PCD produce less CXCL8 upon stimulation with fMLF and LPS compared to PMNs from healthy controls. PMNs from patients with PCD (PCD, *n* = 11–15, 5-7 pediatric (indicated with dark circles) and 6-8 adult (open circles), age-matched healthy pediatric controls (Ped CO, *n* = 4–12) and healthy adult controls (Ad CO, *n*= 11–15) were supplemented with GM-CSF (20 ng/mL) and stimulated with (B,D) LPS (10 µg/mL), (**C**,**E**) PGN (10 µg/mL) or (**A**) fMLF (10^−7^ M) for 24 h at 37 °C. (**A**–**C**) CXCL8 and (**D**,**E**) IL-1β levels in the supernatants were measured with sandwich ELISA. The cytokine levels are expressed relative to the cytokine levels produced by the PMNs of the corresponding reference Ad CO (equal to 100%, not shown). Statistical differences were determined by Mann Whitney U-test (^$^
*p* < 0.05).

**Table 1 ijms-22-06172-t001:** Patient characteristics.

	PCD	Healthy Adult Controls (Ad CO)	Healthy Pediatric Controls (Ped CO)
**Female/Male No.**	21/27 (44%/56%)	16/5 (76%/24%)	23/10 (70%/30%)
**Age, median (range) in years**	14 (3–69)	27.0 (23–44)	11.1 (4–18)
**Pediatric/Adult No.**	32/16 (67%/33%)	0/21 (0%/100%)	33/0 (100%/0%)
**Ultrastructural defect**			
Normal ultrastructure	20/48 (42%)		
ODA deficiency	20/48 (42%)		
IDA deficiency + CP abnormalities	3/48 (6%)		
CP deficiency	4/48 (8%)		
Ciliary aplasia	1/48 (2%)		
**Genetic abnormalities**			
Unknown	8/48 (17%)		
DNAH11	13/48 (27%)		
DNAH5	11/48(23%)		
DNAAF1	3/48 (6%)		
SPAG1	2/48 (4%)		
CCDC39	1/48 (2%)		
CCDC40	2/48 (4%)		
CCDC65	1/48 (2%)		
CCDC103	2/48 (4%)		
HYDIN	1/48 (2%)		
RSPH1	1/48 (2%)		
RSPH4	1/48 (2%)		
RSPH9	1/48 (2%)		
OFD1	1/48 (2%)		
**Clinical features**			
Bronchiectasis	33/48 (69%)		
Situs inversus	17/48 (35%)		
FEV_1_, median (range) (%)	92 (43–113)		
FVC, median (range) (%)	100 (76–125)		
**Laboratory features, median (range)**			
WBC count (/µL)	7415 (3900–17,240)		
CRP (mg/dL)	1.1 (<0.3–10.2)		
**Sputum bacteria**			
*H. influenzae*	10/48 (21%)		
*P. aeruginosa*	7/48 (15%)		
*S. aureus*	10/48 (21%)		
*S. pneumoniae*	4/48 (8%)		
**Treatment**			
Azithromycin	6/48 (13%)		
Oral antibiotics	3/48 (6%)		
Inhaled antibiotics	2/48 (4%)		
Inhaled steroids	1/48 (2%)		

Abbreviations. CP: central pair; CRP: C-reactive protein; FEV_1_: forced expiratory volume after 1 second; FVC: forced vital capacity; IDA: inner dynein arm; ODA: outer dynein arm; WBC: white blood cell

**Table 2 ijms-22-06172-t002:** Overview of peripheral blood neutrophil dysfunction in common chronic neutrophil-dominated lung disorders.

Neutrophil Function	CF	COPD	Asthma	PCD
Phagocytosis	=[46]	=/↓[48]	↑[25,49]	=
ROS production	≠[27,50]	↑[43]	=/↑[25,45,51]	↓
NET formation	↑[24,47]	*NA*	*NA*	↓
CXCL8 production	↑[52,53]	*NA*	=/↑[29,54]	↓
Chemotaxis	≠[55,56,57]	↑[28]	↑[49]	↓
Adhesion molecules	=[41,42]	↑[43,44]	↑[49]	=

Symbols: =, similar to healthy control; ≠, increase or decrease dependent on the stimulus; ↑, increase; ↓, decrease. Abbreviations: CF, cystic fibrosis; COPD, chronic obstructive pulmonary disorder; NA, no data available; NET, neutrophil extracellular trap; PCD, primary ciliary dyskinesia; ROS, reactive oxygen species.

## Data Availability

All data are available on request from the authors.

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
