# Peer review of "The Antimicrobial Activity of Peripheral Blood Neutrophils Is Altered in Patients with Primary Ciliary Dyskinesia"

_ijms, 2021, doi:10.3390/ijms22126172_

Round 1

Reviewer 1 Report

the mechanisms analysed in this study on neutrophil function in patients with ciliary dyskinesia are relevant and very well analysed

Author Response

The mechanisms analysed in this study on neutrophil function in patients with ciliary dyskinesia are relevant and very well analysed

We thank the reviewer for his/her positive evaluation.

Reviewer 2 Report

ijms-1214675 entitled “The Antimicrobial Activity of Peripheral Blood Neutrophils is Altered in Patients with Primary Ciliary Dyskinesia” submitted by Cockx et al.

Authors examined activities of peripheral bood PMNs from PCD patients. The PCD neutrophils produced less reactive oxygen species upon stimulation with bacterial products or cytokines, and the capacity to release DNA. The PMNs from PCD patients do not show over-activation, in contrast to those from FC or COPD patients. The results are potentially interesting. However, there are many unresolved questions.

  1. In PMNs from PCD patients, the ROS production and DNA release stimulated by fMLF or cytokines is reduced. Authors compared with responses in PMNs from PCD patients with those from the healthy control. Are these responses caused by the chronic infections or genetic defects? Although authors cited the results from COPD or CF patents, similar experiments using PMNs from patients having chronic airway infections, such as asthma and COPD patients, are required, to answer the questions.

  1. This manuscript is likely to present the results unadequately. This causes the manuscript to be difficult to follow the results. The reviewer feels to improve this manuscript to follow the results easily.

  1. Authors used many abbreviations in this manuscript. Unfortunately, many of them are not shown the full name, their functions or what they mean. Authors are required to summarize abbreviations according to their functions.

  1. Are these results directly to help PCD diagnosis? However, if PCD genes affect MLN functions, the results are interesting for understanding the MLN activities.

Author Response

We thank the reviewer for his/her concern and evaluation.

  1. In PMNs from PCD patients, the ROS production and DNA release stimulated by fMLF or cytokines is reduced. Authors compared with responses in PMNs from PCD patients with those from the healthy control. Are these responses caused by the chronic infections or genetic defects?

We have added a passage in the discussion (page 15) explaining that the results obtained are most likely a secondary effect of the chronic inflammation, as the mutations in our patient cohort are mostly located in genes that are normally not expressed in the immune compartment.

Although authors cited the results from COPD or CF patients, similar experiments using PMNs from patients having chronic airway infections, such as asthma and COPD patients, are required, to answer the questions.

We agree that the inclusion of disease controls would be a valuable addition to our study. However, we believe that it is very hard to identify reliable disease controls. In cystic fibrosis, there is clear evidence of neutrophilic dysfunction, partially caused by the genetic defect in CFTR, expressed in the neutrophil itself.

COPD is a non-genetic disease that is mainly induced by smoking, and is very heterogeneous. Previous data have shown neutrophilic abnormalities, although it is not clear what the origin of these is.

Non-CF bronchiectasis is a very heterogeneous group of disorders that is characterized by bronchiectasis and chronic neutrophilic endobronchial infection. However, the etiology of this disease varies and is difficult to identify in most patients. The clinical spectrum is wide with a big range in disease severity. Moreover, it is not unthinkable that a primary neutrophilic problem could be the origin of the disease in some patients.

We included a table summarizing the current evidence in the literature on neutrophil abnormalities in CF, COPD and asthma in the discussion of the paper (Table 2, page 15).

  1. This manuscript is likely to present the results inadequately. This causes the manuscript to be difficult to follow the results. The reviewer feels to improve this manuscript to follow the results easily.

We feel sorry that the reviewer finds it difficult to follow the results. In this study, we have followed the standard article structure as recommended by MDPI. Other reviewers did not indicate any difficulties in following the flow of the results.

  1. Authors used many abbreviations in this manuscript. Unfortunately, many of them are not shown the full name, their functions or what they mean. Authors are required to summarize abbreviations according to their functions.

We have added a list of abbreviations at the end of the article.

  1. Are these results directly to help PCD diagnosis? However, if PCD affects PMN functions, the results are interesting for understanding the PMN activities.

We have added a passage in the discussion (page 14) highlighting the relevance of our results. Our data mainly show a reduced activation of peripheral blood neutrophils in PCD, as opposed to those found in CF, COPD and asthma. It is not yet clear whether these findings could be applied to facilitate the diagnosis of PCD; however, they may suggest that the administration of neutrophil-suppressing drugs will not be beneficial in these patients. 

Reviewer 3 Report

The Authors investigated thoroughly the functionality of polymorphonuclear cells (PMN) in patients affected by Primary Ciliary Dyskinesia (PCD), a disease where these cells play a crucial role by sustaining  an inflammatory state  and killing bacterial pathogens. They analyzed several processes in PMN cells from PCD patients comparing them to PMN cells from healthy controls and found:

- a reduced chemotaxis but unaltered expression of fMLF receptors

-normal expression of adhesion molecules

-no phagocytic defects

-a reduced production of reactive oxygen species upon stimulation

-a diminished ability to release DNA during extracellular trap formation.

The work is appropriately designed and results are well discussed. Nevertheless, the manuscript needs some minor revisions, especially for what concerns table 1:

the table reports some numerical discrepancies relative to the text (i.e. the total number of PCD patients indicated in the top raw of the table is 46 and not 48 as indicated on page 2, line 93, and in the rest of the table. In addition, the total number of ultrastructural features does not match the total of patients).

The distinction between adult and pediatric individuals made for healthy controls should also apply to PCD patients. This is particularly important, because in PCD the prevalence of some pathological features, such as bronchiectasis or presence of bacterial pathogens, is age-dependent.

It would be useful to add in the table the relative prevalence (%) of the ultrastructural defects and sputum bacteria.

Other points that should be addressed:

  • Has genetic testing been performed in PCD patients?
  • If the patients were clinically stable, why they were taking antibiotics?
  • I have not found in the text the citation number 47.
  • At the beginning of the reference section there is a typing error and the paragraph should be deleted.

Author Response

  1. The table reports some numerical discrepancies relative to the text (i.e. the total number of PCD patients indicated in the top row of the table is 46 and not 48 as indicated on page 2, line 93, and in the rest of the table. In addition, the total number of ultrastructural features does not match the total of patients). The distinction between adult and pediatric individuals made for healthy controls should also apply to PCD patients. This is particularly important, because in PCD the prevalence of some pathological features, such as bronchiectasis or presence of bacterial pathogens, is age-dependent. It would be useful to add in the table the relative prevalence (%) of the ultrastructural defects and sputum bacteria.

Other points that should be addressed:

  • Has genetic testing been performed in PCD patients?
  • If the patients were clinically stable, why they were taking antibiotics?
  • I have not found in the text the citation number 47.
  • At the beginning of the reference section there is a typing error and the paragraph should be deleted.

We thank the reviewer for his/her concern and evaluation. We have re-evaluated the table (Table 1) and added the necessary information. The information about the number of pediatric and adult patients has also been added to the figure legends.

We have deleted the typing error at the beginning of the reference section.

Concerning the oral antibiotic intake, indeed a minority of patients received maintenance therapy with oral antibiotics to reduce the frequency of exacerbations and to obtain stable status. 

Round 2

Reviewer 2 Report

I have enjoyed the manuscript reading.